# On the Critical Condition for Flame Acceleration in Hydrogen-Based Mixtures

**DOI:** 10.3390/ma16072813

**Published:** 2023-03-31

**Authors:** Alexey Kiverin, Alexey Tyurnin, Ivan Yakovenko

**Affiliations:** Joint Institute for High Temperatures of Russian Academy of Sciences, Izhorskaya st. 13 Bd.2, Moscow 125412, Russia

**Keywords:** hydrogen, hydrogen energy, hydrogen combustion, hydrogen safety, propulsion, flame acceleration, numerical analysis

## Abstract

The paper presents a novel numerical approach to the quantitative estimation of the concentration limits for flame acceleration in hydrogen-based mixtures. A series of calculations are carried out for hydrogen–air and hydrogen–oxygen flames in channels. The analysis of the obtained numerical results provided the value of 11 ± 0.25 % hydrogen content in the mixture as a lean concentration limit of flame acceleration that agrees well with the available experimental data. Moreover, the basic physical mechanism responsible for the transition from the steady mode of flame propagation to the accelerated one is distinguished. The mechanism is related to flame stretching in the region of interaction with the boundary layer and the competition between the joint increase in burning rate and heat losses. The novel technique for the estimation of concentration limits of flame acceleration presented here can be applied to assess combustion conditions inside combustors of energy and propulsion systems fed with hydrogen. The results are also useful in estimating explosion and fire risks in hydrogen storage, transport, and utilization facilities as parts of hydrogen energy and propulsion systems.

## 1. Introduction

Hydrogen is among the prospective substances to be used as a gaseous fuel for propulsion and energy, including green energy. It can be utilized as a fuel for engines by itself [1] or as an additive to conventional types of fuels [2]. Moreover, hydrogen is an energy carrier for fuel cells, which can also be used for propulsion [3,4]. The application of hydrogen as a fuel faces numerous challenges concerning its safe storage, transfer, and efficient utilization. Moreover, hydrogen safety problems are also of great importance since hydrogen is one of the most hazardous gaseous fuels, characterized, in particular, by its wide flammability limits [5] and low ignition energy [6].

As low-density gaseous hydrogen is usually stored under high pressure inside a vessel, the risk of vessel damage defines the risk of hydrogen release and self-ignition [7]. There are many factors affecting the probability of self-ignition [8,9], including the storage pressure value, the rate of hydrogen release, the composition of the atmosphere surrounding the vessel, the geometry of the surrounding space, etc. At relatively low rates of hydrogen release (insufficient for hydrogen self-ignition), the surrounding space is filled up with hydrogen as it mixes with the gaseous atmosphere and represents an explosive medium. In these conditions, any accidental local energy release could induce explosion development. Therefore, one of the main goals of the theory of combustion and explosion is to define the critical conditions at which different modes of explosion development can occur.

The particular mode of explosion development depends on numerous factors such as mixture composition, space geometry, and gas-dynamic flows, including turbulent flows. Thus, the following consequences of explosion modes occur with the increase in hydrogen content in the mixture of an oxygen-containing atmosphere. At low levels of hydrogen in the mixture, there is no explosion at all, and ignition can only take place when the hydrogen content achieves a certain critical value called the “lean flammability limit” [5]. Furthermore, there is a region of so-called ultra-lean flames [10,11]. Here, the explosion develops in the form of isobaric “flameballs”, which are affected by the convective flows determining the motion of such “flameballs” inside the vessel [10,11]. Under microgravity conditions, such “flameballs” appear to be stable and do not propagate in space until all the hydrogen is burned down [12]. Thus, the value of the lean flammability limit differs from the one in the terrestrial conditions, and ultra-lean “flameballs” can emerge under micro-gravity conditions at lower hydrogen content (∼3% compared to ∼4% in terrestrial conditions). The turbulization of the external flow can affect the lean flammability limit [13]. Further increases in the hydrogen content lead to a mode of combustion development called a deflagration wave (a classic combustion wave). A deflagration wave is also sensitive to the mixture composition as well as to different external factors. In particular, the deflagration wave can accelerate, which is one more mode of hydrogen explosion development, and under certain conditions, the deflagration can transit into detonation [14]. Given this, there are certain critical conditions for deflagration acceleration, transition to detonation, and stable detonation propagation to occur [15].

When considering hydrogen as a fuel for propulsion systems, the critical conditions of stable combustion are of primary importance [16,17]. The elaboration of conceptual devices utilizing the detonation mode of combustion [18] also demands knowledge about the conditions for transition to detonation and its stable propagation. Thus, flame acceleration usually is treated as the main reason for the deflagration-to-detonation transition [15]. Moreover, the accelerated deflagration wave can be also efficiently used for propulsion, providing close parameters to the detonation mode [19].

As both the safe and efficient use of hydrogen in propulsion is in demand, it is of great interest to establish the critical conditions for flame acceleration. To our knowledge, there are no papers where a numerical approach for the quantitative estimation of the minimum hydrogen concentration required for the flame acceleration process is provided. So, the main goal of this paper is to propose such an approach to the quantitative estimation of the concentration limits for the realization of the accelerated flame mode. The study is carried out with the help of numerical simulations, and the obtained results agree well with the known experimental facts. Furthermore, the proposed approach for quantitative estimation of concentration limits can be used in the assessment of combustion conditions inside the combustors of energy and propulsion systems. It should be noted that based on the carried out calculations, the analysis of the mechanism of flame acceleration is also presented in the paper. That, together with the formulated approach, constitutes the basis for the assessment of risks related to a contingent hydrogen explosion.

## 2. Problem Setup

Recently in [20], it was proposed that there should be similarity in the flame development in channels of various geometries filled with different mixtures. This hypothesis was checked on numerical data on the flames propagating inside channels of different widths filled with a stoichiometric hydrogen–air mixture. It was shown that at least the initial stages of flame evolution are similar in different channels, and one can use scales normalized by channel width (such as x/H or $u_{f}t/H$) when analyzing the dynamics of the flame. Furthermore, this was also justified experimentally in [21]. According to this, one can use direct numerical simulations of flame propagation through a relatively narrow channel to obtain information on whether the flame can accelerate or not. However, it should be noted that the heat losses responsible for preventing flame acceleration in narrow channels can be neglected in the framework of the numerical analysis. Therefore, this paper proposes to look for the critical concentration of hydrogen needed for flame acceleration via a series of calculations related to flame propagation in a narrow channel filled with different hydrogen-based mixtures.

The schematic of the problem setup is shown in Figure 1. The semi-opened channel of width H and length 20 H–70 H is initially filled with a quiescent mixture of the given composition under normal conditions (300 K, 1 atm). The flame is initiated in the vicinity of the closed end of the channel inside a small preheated region of a 1 mm radius. The mixture inside the ignition zone is instantaneously heated to 1500 K at constant pressure. The channel walls are non-slip, and the adiabatic condition is additionally applied to reduce the effect of heat losses on the channel walls. As it is mentioned above, in the absence of heat being lost to the walls, it becomes possible to distinguish the gas-dynamic mechanisms responsible for flame development in channel geometry, independent of the channel width. As a result, it is assumed that such calculations can become a basis for an approach to the estimation of concentration limits of flame acceleration. Outflow conditions on the far right boundary of the computational domain are calculated according to the relationship between the characteristics connecting the boundary region with the far region, where the medium is quiescent and is at normal conditions.

Two series of calculations are carried out for hydrogen–air and hydrogen–oxygen mixtures with the hydrogen content varying from 10% to 29.5%. In particular, we start from the case of a quite rich mixture with, e.g., 17% of hydrogen, and then decrease the hydrogen content with a step of 1%. After reaching a state without flame acceleration (e.g., 11%), we go back with a step of 0.25% to determine the critical hydrogen content with an accuracy of 0.25%. When analyzing the numerical results, the flame speed is chosen as the main parameter describing the evolution of the process and is defined as the speed of the leading point of the flame front.

The calculations are carried out based on the conventional gas-dynamic model for reactive gas [22]. The mathematical model is represented by the full Navier–Stokes equations written in two-dimensional Cartesian coordinates. In addition, all processes, such as compressibility, viscosity, heat conductivity, multi-component diffusion, and chemical reactions, are taken into account. The only neglected process is radiant heat transfer. It should be noted, however, that both the combustion products and the fresh mixture can be considered as optically transparent media, and the walls are set up to be adiabatic. Therefore, the effect of radiation on the flame development is assumed to be negligible. Hydrogen oxidation is modeled with the use of a detailed kinetic mechanism from [23]. That kinetic mechanism includes 19 reactions for 8 species, namely $H_{2}$, $O_{2}$, H, O, OH, ${\mathrm{HO}}_{2}$, $H_{2}O_{2}$, and $H_{2}O$. Nitrogen takes part in the reactions only as a third body, so its oxidation is not taken into account. The novel second-order explicit high-resolution numerical technique CABARET [24] is used to perform calculations. Auxiliary variables (flux variables) defined on the staggered grid in space and in time are introduced in CABARET through upwind extrapolation, which determines the number of benefits of this technique. CABARET is characterized by very its small dissipation and dispersion errors, its parameter-free flux correction procedure based directly on the maximum principle of the flux variables, its compact computational stencil leading to low computational cost per time iteration, and its effective implementation for parallel computing via the domain decomposition approach. The characteristic decomposition method underlying the CABARET technique allows for a straightforward description of the inflow and outflow boundary conditions via the relationships between characteristics. Recently, this numerical technique was adopted by several authors for the numerical analysis of combustible systems, and they showed its usefulness in solving the problems of reactive gas dynamics [25,26,27]. In particular, the chosen numerical technique is thoroughly validated for hydrogen–air flames in [25]. Moreover, it reproduces the non-steady effects in flame development well and predicts with high accuracy such a phenomenon as the deflagration-to-detonation transition [27] (the calculations agree well with the available experimental data). Although a numerical technique with low dissipation is used, one should resolve the reaction zone with high accuracy. So, the computational grid resolution is chosen in such a way so as to adequately resolve the flame front’s inner structure. For example, in the case of a near-limit mixture containing 11% of hydrogen, the flame thickness is of the order of 1.3 mm, which is resolved well with a computational grid with a cell size of 0.1 mm. When carrying out the convergence tests, it is controlled so that the convergence rate estimated according to Richardson’s technique [28] is not lower than unity; in the ideal case, it aims to be 2.0, which corresponds to the order of accuracy of the numerical technique. Moreover, it is preliminarily assumed that the numerical prediction should be predicted with an error no higher than 10-15% compared with the estimate for the exact solution. At the same time, such an error is sufficient for two-dimensional calculations, which are quite demanding in terms of computational resources. Figure 2 shows the characteristic dependence of the flame thickness on the linear size of the numerical cell (δx) in the example of flame propagation through an 11% hydrogen–air mixture. In this particular case, the rate of convergence at δx=0.1 mm is equal to 2.0, which corresponds to its maximum when using a second-order numerical technique. Thus, the error in the prediction of the flame structure is estimated as 2%.

## 3. Results and Discussion

Let us first consider the process of flame propagation in the channel. The process starts after successful ignition in the localized preheated area (Figure 1), and as soon as the flame front is formed, it propagates isotropically as an almost ideally spherical flame. Two main mechanisms are involved in this stage: the expansion of hot combustion products and the gas-dynamic instability of the outwardly propagating flame. Both mechanisms define flame acceleration from the value of normal burning velocity $u_{f}$ to the value of $\Theta u_{f}$, where Θ is the expansion ratio, which is the ratio of the densities of the fresh mixture and combustion products, and further to higher values via the mechanism of instability development [29]. The expansion of combustion products defines the compression and motion of the fresh mixture ahead of the flame front in the direction of the channel walls. As soon as the compressed moving gas interacts with the wall, the compression wave reflects from the wall and propagates towards the flame front, affecting its further development. In particular, the deceleration of the flame takes place, and the overall shape of the flame becomes elongated along the walls (Figure 3) since there is no deceleration effect from the far end of the channel. Such a flame is called a “finger flame” [30], and it propagates with an acceleration induced by the positive feedback between the flame speed and the increase in flow velocity [31]. The speed of the "finger flame" increases exponentially over time (Figure 4, stage “I”) because of the positive feedback mechanism. However, this stage of flame acceleration is only an intermediate one and is limited in time.

When the flame propagates out from the closed-end wall with relatively low velocity, one can observe a pressure equalization phenomenon in the region behind the flame front. This is primarily related to the fact that the rarefaction of the combustion products proceeds almost instantly in the background of subsonic flame propagation. However, when the front of the “finger flame” propagates far from the closed-end wall, the rarefaction of the combustion products takes more time. So, the rarefaction region emerges behind the flame front and moves asymptotically to the case of flame propagation out from the open end [32]. That rarefaction incorporates gas into the motion counter to the direction of the flame propagation. As a result, the flow decelerates, as does the flame (Figure 4, stage “II”). Here, it should be noted that this effect is stronger in the bulk flow, while the motion near the side walls remains almost unaffected. Due to this, the most drastic flame deceleration is observed in the central part of the channel, and the “finger flame” transforms into the so-called “tulip flame” (Figure 3). Further dynamics of the flame is defined mainly by the interaction of the formed “tulip flame” with the boundary layer developing near the side wall of the channel. Under certain conditions, such an interaction leads to the flame stretching along the walls, coupled with flame acceleration. Thus, for example, one can clearly see that in the cases presented in Figure 3a–c, the “tulip flame” elongates after its formation (last profiles in Figure 3). In such a case, a certain flame acceleration occurs (Figure 4, 12% and 17% hydrogen–air mixtures). At the same time, there is no such effect in less reactive mixtures (Figure 3d and Figure 4, 11 %); in such a case, the flame shape, as well as the flame speed, remains the same, and no notable flame elongation is observed. The increase in pressure represents an important sequence of flame acceleration in a channel. As Figure 4c clearly shows, there is an unambiguous dependence between the flame acceleration and the increase in pressure. Given this, the pressure measurements can also be fruitfully used when distinguishing the modes of flame propagation with and without acceleration.

Flame stretching in the gas flow ambiguously affects the combustion development. On the one hand, the increase in the flame area defines the increase in the volume of the fresh mixture burned down per time unit. So, there is a certain increase in the flame speed. On the other hand, a local increase in the flame front area defines the increase in the heat losses from the stretched part of the flame front, which could even lead to local flame quenching in the case of less reactive mixtures. In particular, that mechanism is responsible for the quenching of both laminar [11] and turbulent [33,34] flames. As a result, when flame stretching occurs due to the flame’s interaction with the boundary layer, competition between those two mechanisms arises in the flame stretching region. In the case of chemically active mixtures, the energy released in the reaction zone is sufficient to compensate for losses. Therefore, the stretching leads to flame acceleration. Under such conditions, the “tulip flame” propagates with permanent acceleration. In less chemically reactive mixtures, additional losses induced by flame stretching, together with the lower energy release, leads to combustion stabilization. In that case, the “tulip flame” is not accelerating, and a quasi-steady mode is established.

Based on the series of calculations carried out, one can determine the mixture composition corresponding to the transition from the mode of flame acceleration to the mode of quasi-steady flame propagation. Thus, for example, in the case of hydrogen–air mixtures (see Figure 3 and Figure 4), such concentration limit equals 11.25 ± 0.25%. It is important to note that the obtained critical value agrees well with the available experimental data [35,36]. According to the correlation obtained in [35] based on the series of experiments, ${\left[H_{2}\right]}_{cr}=11.07$ %. Moreover, the same lean concentration limit (11.25 ± 0.25% of hydrogen in the mixture) for the mode of flame acceleration is obtained for hydrogen–oxygen mixtures. This result also agrees well with the experimental data [37], according to which the concentration limit of flame acceleration in $H_{2}$/$O_{2}$/$N_{2}$ mixtures at normal conditions is always somewhere in the vicinity of 11% hydrogen content in the mixture. This effect can be explained as follows. Here, quite lean mixtures are considered, so both nitrogen and excess oxygen represent neutral components of the mixture. Due to this, the replacement of oxygen with nitrogen has almost no effect on the burning rate and flame temperature. Thus, numerical analysis shows that the flame temperature rises by no more than 0.5–1.0% when replacing the air with oxygen (at 11% of hydrogen in the mixture). Thus, the rates of heat release in the reaction zone and heat transfer from the reaction zone to the fresh mixture are much more sensitive to the changes in the composition of the oxidizer. Thus, Figure 5 illustrates the changes in characteristic time scales defining the burning rate ($\tau_{b}=L_{f}/u_{f}$, where $L_{f}$ is the flame front thickness) and the heat transfer on the scales of the flame front ($\tau_{\chi }=L_{f}^{2}/\chi$). One can clearly see that both parameters change with the replacement of air with oxygen. However, at the same time, those changes are of the same order. Therefore, the competitive mechanism between the heat release and heat losses under the flame stretching described above is quantitatively the same in both the hydrogen–oxygen and hydrogen–air mixtures. That is why the concentration limit of flame acceleration remains the same, independent of the nitrogen content in the oxidizer.

Note that the obtained critical mixture composition containing ∼11 % hydrogen in air is close to the limit at which the mechanism of flame propagation is changed from the deflagration mode to a significantly different one defined mainly by the diffusion of hydrogen into the reaction zone [34]. Such combustion modes are also affected by the gas dynamics (as it is clearly demonstrated in our previous works, e.g., in [10]), but there is a certain flame speed limit that can be achieved, and one should not expect further flame acceleration after that speed limit is achieved. Without a doubt, the question arises if the concentration limit for flame acceleration obtained here is the same as the concentration limit for the existence of deflagration, but this question should be resolved separately elsewhere. From our point of view, this particular question deserves to be highlighted as one of the results of this particular work.

Let us also discuss the effect of the increased pressure in the process of flame propagation. As is mentioned above, when analyzing the data presented in Figure 4, the increase in pressure in the semi-opened channel can be observed only in the case of flame acceleration. At the same time, one should not expect a notable increase in pressure in the case of a quasi-steady mode of flame propagation. Therefore, as one can clearly see, the rate of increase in the pressure in the near-limit mixtures is quite slow, so one should not expect the generation of strong shock waves, at least at the early stage of flame development considered here. Meanwhile, the pressure is the principal factor for the deflagration-to-detonation transition, and it is well known that such a transition can take place only when significant rates of acceleration and certain rates of pressure build-up are achieved. Due to this, the concentration limits of the deflagration-to-detonation transition are much narrower compared with the limits of flame acceleration (see, e.g., [36]). Nevertheless, the “weak” flame acceleration (in lean mixtures) on quite large scales and in the presence of obstacles can lead to quite high flame speeds and overpressure, which should be treated as a hazardous factor or, alternatively, can be fruitfully used in propulsion.

## 4. Conclusions

In the present paper, the numerical analysis of flame propagation through smooth channels filled with hydrogen–air and hydrogen–oxygen mixtures under normal conditions is carried out. It is shown that there is a certain critical hydrogen content in the mixture corresponding to the transition between the mode of flame acceleration and the quasi-steady mode. Thus, flame acceleration becomes possible only at a hydrogen content higher than 11.25 ± 0.25%. As a result of numerical analysis, the basic physical mechanism responsible for the transition is distinguished. That mechanism is defined by the competition between the local acceleration of burning and the local intensification of heat losses from the reaction zone to the fresh mixture when the flame is stretched in the region where it interacts with the boundary layer. It is important to note that the obtained numerical results on the lean concentration limit of hydrogen flame acceleration agree well with the available experimental data, leading to the conclusion that the numerical setup proposed in this paper can be used as a basis for an approach to the estimation of critical conditions for flame acceleration. According to this approach, one can estimate certain concentration limits at a given initial thermodynamic state of the mixture based on the series of calculations carried out in relatively narrow channels. The proposed approach, combined with the use of numerical simulations, can be widely used to predict the real conditions in which flame acceleration can take place. Such a an approach can be quite useful in the assessment of combustion conditions inside combustors of energy and propulsion systems fed with hydrogen and when estimating hazardous risks related to hydrogen leakage and subsequent explosion. Note that the current results are applicable only for confined volumes, while flame acceleration in free space develops according to distinct mechanisms and, therefore, should be considered separately [27].

Among our future goals, it is important to figure out if the concentration limit for flame acceleration obtained here is the same as the concentration limit for deflagration to occur. Furthermore, much more experimental and numerical data for different combustible mixtures are needed to obtain a clear understanding of the range in which the proposed routine can be used.

## Figures and Tables

**Figure 1 materials-16-02813-f001:**
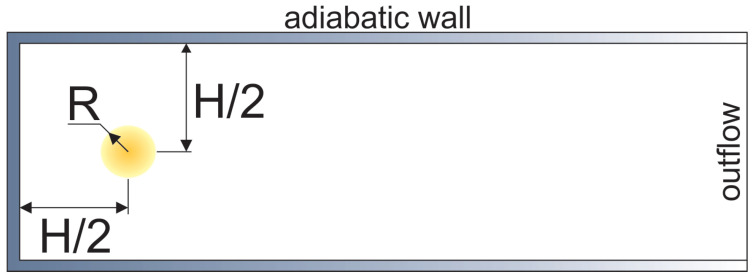
Schematic of the problem setup. Channel width equal to H; in most of calculations H = 10 mm, R = 1 mm.

**Figure 2 materials-16-02813-f002:**
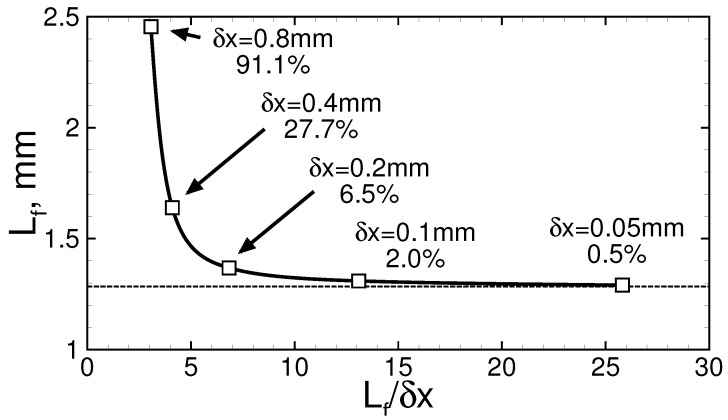
Results of the convergence test for 11% hydrogen in air combustion. The curve with signs shows the flame thickness’s dependence on the linear size of the numerical cell. The dashed line shows the exact solution estimated according to Richardson’s technique. Numerals show the sizes of the numerical cell and the error value in percent.

**Figure 3 materials-16-02813-f003:**
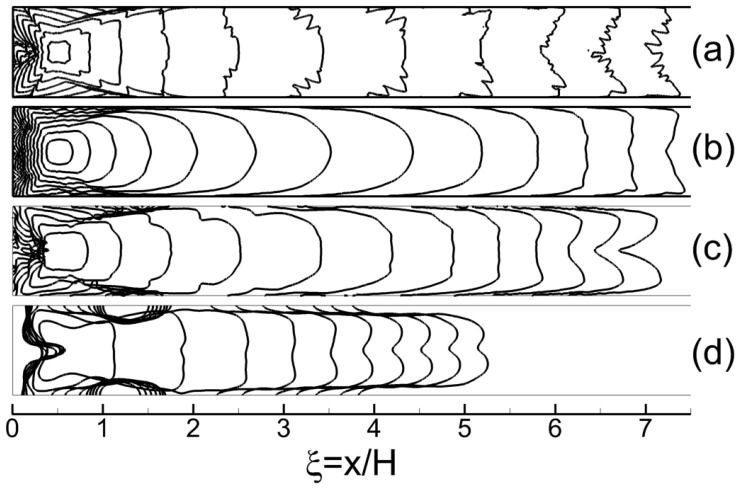
Evolution of the flame in channel: (**a**) H = 40 mm, 29.5 % of hydrogen in mixture; (**b**) H = 10 mm, 29.5 %; (**c**) H = 10 mm, 17.0 %; (**d**) H = 10 mm, 11.0 %. Flame position is shown with the use of temperature iso-lines T = 1000 K. Time intervals between subsequent instants $u_{f}\Delta t/H\approx 0.025$ (400 μs (**a**), 100 μs (**b**), 500 μs (**c**), 3750 μs (**d**)).

**Figure 4 materials-16-02813-f004:**
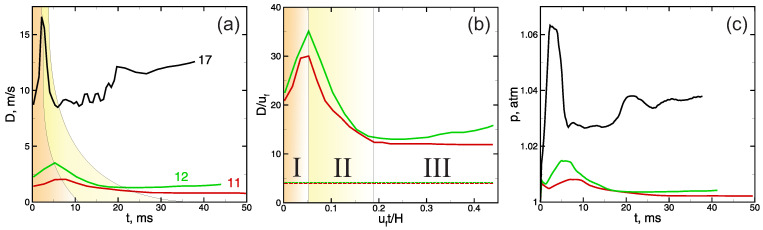
Time histories of flame speeds (**a**,**b**) and maximal pressure in channel with H = 10 mm filled with different mixtures containing 11 % (red), 12 % (green), and 17 % (black) hydrogen. Highlighted regions correspond to the early stage of flame acceleration (I), stage of flame deceleration (II), and further flame development either with acceleration or propagation with a constant speed. Flame speed (D) and time (t) are presented in SI (**a**,**c**) and in dimensionless values (**b**). Pressure (p) is presented in atm units. Dashed lines in frame (b) show the values of $D=\Theta u_{f}$, while $D/u_{f}=1$ corresponds to the value $D=u_{f}$.

**Figure 5 materials-16-02813-f005:**
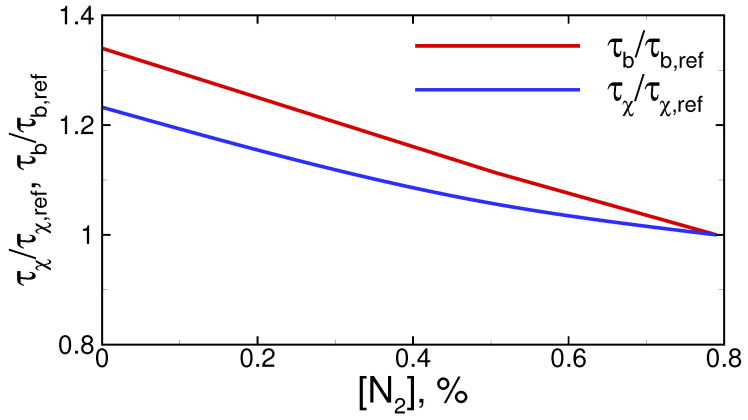
Time scales characterizing the energy release inside the reaction zone ($\tau_{b}$) and heat losses due to heat transfer ($\tau_{\chi }$) depending on the nitrogen content in the oxidizer (O${}_{2}$/N${}_{2}$). Hydrogen content in all the mixtures is 11 %. Values with index “ref” correspond to the hydrogen–air mixture ($\left[N_{2}\right]$ = 79 %).

## Data Availability

The datasets generated and/or analysed during the current study are available from the corresponding author on reasonable request.

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
