# Peer review of "On the Critical Condition for Flame Acceleration in Hydrogen-Based Mixtures"

_materials, 2023, doi:10.3390/ma16072813_

Round 1

Reviewer 1 Report

Thank you for the effort to prepare your paper.

Please consider the following comments.

- The reference [23] considers the reaction mechanism for hydrogen and syngas mixture. Please note the correct reaction mechanism used in the paper.

- In this paper, only the channel geometry was considered. Is the result  (11.25% H2 concentration) applicable to open-boundary? Please discuss the applicability of the results.

- It is known that thermal radiation affects on the hydrogen flame propagation. But in this paper the radiation was not considered. Please discuss the radiation effect on the minimum H2 concentration for the flame acceleration.

- In the research, channel walls are modeled as adiabatic. It is ambiguous what the heat loss means. Please include the criterion of the heat loss.

- This paper shows just a few cases. It is interesting how to find the min. concentration for the flame acceleration. Please  describe the number of cases simulated. It is recommended to  add some figures or tables to show tendency of flame propagation charactericstic depending on the H2 concentration. 

- In the conclusions (line 174), What is the proposed technique? This paper only specifies the used code of CABARET.  Are there any differences or progresses from the referenced CABARET code?  The reference [24] only shows the accuracy of hyperbolic characteristic such as moving waves. Viscous, thermal and mass diffusion are well validated?

Author Response

Dear Reviewer,

We thank you for detailed analysis of our Manuscript and fruitful comments. We revised the Manuscript according to your comments and provide answers in the attached file.

With best regards,

A. Kiverin, A. Tyurnin, I. Yakovenko

Reviewer 2 Report

The manuscript concerns a numerical study of hydrogen flames within channels. The numerical method is not described in detail (combustion models, turbulence models, wall-treatment, numerical schemes, grid-independence analysis). Furthermore, no validation against experimental data is given.  With the information provided, the results cannot be reproduced and their accuracy has not been demonstrated.

Two main issues must be considered:   - the numerical method should be described in detail in order for the simulations to be reproduced by other research group;   - the numerical results should be validated against experimental measurements.   Without these two conditions, the numerical analyses cannot be considered accurate from a scientific point of view.   The topic of the manuscript could be of interest for the scientific community, but the authors have already published many works on the subject (see the self-citations in the References) and the original contribution of this manuscript (concentration limit of H2 in the mixture for flame acceleration in channels) is marginal to justify the publication in a scientific journal. The authors claim " a novel numerical approach for quantitative estimation of the concentration limits for flame acceleration in hydrogen-based mixtures." but DNS surely cannot be considered a "novel approach". Besides, a lot of data could be obtained from DNS simulations, but only 3 figures are provided in the manuscript.

Finally, the original contribution of this manuscript to the scientific literature and to previous works published by the same authors appears marginal.

Author Response

(The authors gave the same response as above.)

Reviewer 3 Report

Manuscript contains the principal new result: to my best knowledge the steady
flame propagation in the semi infinite channel or tube from the closed end
is reported for first time. The solution is obtained be CFD methods.
I congratulate the authors!
But as each new results it demands the extra cushion to numerical procedure
employed. Unfortunately authors very skimpily describe the research.

Let's start from "Problem setup"

Ps1) "small preheated region of a 1 mm radius"

Its temperature, density, pressure and composition are not defined.

Ps2) "All the channel walls are non-slip, and the adiabatic condition is
additionally applied to neglect the effect of heat losses."

a) Does "outflow" wall obeys to "the adiabatic condition"?
b) How long is the channel?
c) "hydrogen content varied from 10% to 30%" Why the results for 10% do not
include into the manuscript?

"Results and discussion"

Rd1) There are several definitions of the speed of the flame. Which one do you
employ?

Rd2) Please, mark $u_f$'s and $\Theta u_f$'s as a points on Fig. 3

Rd3) Please, prolong red line on Fig. 3. In my opinion it starts to go down.

Rd4) "Figure 3, stage “II”", Could you please plot the pressure and the X
component of the gas velocity in laboratory coordinate system for y=H/2 and
y=H/4 for H=10 mm and 11.0% of hydrogen (Fig.2 (d)).

Rd5) "On the other hand, a local increase in the flame front area defines the
increase in the heat losses"

a) please, explain the origin of the "heat losses".
b) Does this statement contradict to Ps2?

Rd6) "In that case, the “tulip flame” is not accelerating, and a quasi-steady
mode is established."

Constant speed of the flame propagation is not enough for "a quasi-steady
mode". Which other parameters of the flame authors check and find quasi-steady?

Rd7) How authors can explain the statements "the heat losses on the scales of
the flame front (τ_\χi = L^2_f/\χi)" ?

Co1) "flame acceleration becomes possible only at a hydrogen content higher
than 11.25 ± 0.25%.

This conclusion is not supported by presented results. For 10% hydrogen mixture
the short interval when "the “tulip flame” is not accelerating," was
demonstrated. Maybe the flame starts accelerate a little bit later(?), maybe
DDT will occurs with no noticeable acceleration (?), maybe flame extincts(?).
Last case means that authors have found the ignition boundary for a parameter
set of the hot spot studied, and another parameters may impact of the ignition.

Author Response

(The authors gave the same response as above.)

Round 2

Reviewer 1 Report

The revised manuscript is thought to have reflected the reviewer's comments well.

So, it is recommended to be published in this journal with an addition of future work because it is thought the research subject is not yet complete.

Author Response

Dear Reviewer #1,

We thank you for detailed analysis of our Manuscript and overall positive estimation of the work. We revised the Manuscript according to your comments and provide answers below.

With best regards,

A. Kiverin, A. Tyurnin, I. Yakovenko

Comment 1: So, it is recommended to be published in this journal with an addition of future work because it is thought the research subject is not yet complete.

Answer 1: We added the comment about the future work in the revised text.

Quality of English Language: English very difficult to understand/incomprehensible.

Answer: We believe that this decision is a misprint since there are no comments and suggestions in the review and there was opposite decision during the first round “English language and style are fine/minor spell check required”.

Reviewer 3 Report

1. The manuscript points to the single condition. Please correct the title:
"conditions" -> "condition".

2. line 132 "convergence tests" Please include the convergence test: parameters
and full results into manuscript.

3. As I can see the convergence rate is close to 1, but you state that you
numerical method is the "second-order ... high-resolution" (line 113).
   a) What does in mean "high-resolution"? Second-order is not high.
   b) Why the second-order method demonstrate the first order convergence?
      Please include your comment into the manuscript.

Author Response

Dear Reviewer #2,

We thank you for detailed analysis of our Manuscript and overall positive estimation of the work. We revised the Manuscript according to your comments and provide answers below.

With best regards,

A. Kiverin, A. Tyurnin, I. Yakovenko

Comment 1: The manuscript points to the single condition. Please correct the title:

"conditions" -> "condition".

Answer 1: We agree. The title is changed.

Comment 2: Line 132 "convergence tests" Please include the convergence test: parameters

and full results into manuscript.

Answer 2: We added an example of such a test for the case mentioned in the previous version of the Manuscript (11%).

Comment 3: As I can see the convergence rate is close to 1, but you state that you

numerical method is the "second-order ... high-resolution" (line 113).

  1. a) What does in mean "high-resolution"? Second-order is not high.
  2. b) Why the second-order method demonstrate the first order convergence?

      Please include your comment into the manuscript..

Answer 3:

  1. Here high-resolution means the ability of the algorithm to reproduce fine details of the flow, including shock and acoustic waves, that is achieved due to the low-dissipation features instead of increased accuracy order.
  2. We added an example of a convergence test for the case mentioned in the previous version of the Manuscript (11%) and the convergence rate equals to 2. Previously we just mentioned that we controlled this value not to be lower than unity. It is known that the convergence rate estimated according to the Richardson’s routine is always lower than the order of the numerical scheme if there is no super convergence due to some elements of the model, e.g. boundary conditions, sources etc.
